# EXACT: A Video-Language Benchmark for Expert Action Analysis

**Han Yi    Yulu Pan    Feihong He    Xinyu Liu    Benjamin Zhang**
**Oluwatumininu Oguntola    Gedas Bertasius**
University of North Carolina at Chapel Hill
{hanyi, yulupan, gedas}@cs.unc.edu, {feihonhe, xinyu1, zhangben, oguntola}@unc.edu

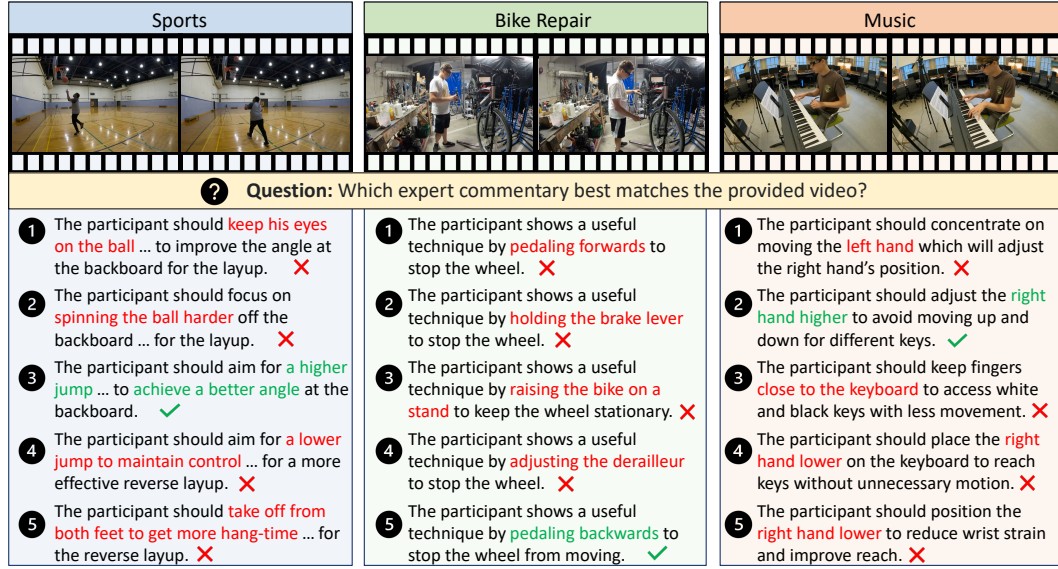

Figure 1: An illustration of several multiple-choice question samples from our expert action analysis benchmark, EXACT. Here, we visualize samples from three domains of skilled activities (i.e., basketball, bike repair, and piano). The correct answers have a green checkmark next to them and were obtained using domain-expert/coach annotations. The phrases in green (correct) and red (incorrect) emphasize the subtle yet critical differences between ground-truth expert descriptions and incorrect candidate answers.

## Abstract

We present EXACT, a new video-language benchmark for expert-level understanding of skilled physical human activities. Our new benchmark contains 3,521 expert-curated video question-answer pairs spanning 11 physical activities in 6 domains: Sports, Bike Repair, Cooking, Health, Music, and Dance. EXACT requires the correct answer to be selected from five carefully designed candidate options, thus necessitating a nuanced, fine-grained, expert-level understanding of physical human skills. Evaluating the recent state-of-the-art VLMs on EXACT reveals a substantial performance gap relative to human expert performance. Specifically, the best-performing Gemini 2.5 Pro model achieves only 55.35% accuracy, well below the 82.02% attained by trained human experts. We believe that EXACT will be beneficial for developing and evaluating VLMs capable of precise understanding of human skills in various physical and procedural domains. Dataset and code are available at https://texaser.github.io/exact_project_page/.

39th Conference on Neural Information Processing Systems (NeurIPS 2025) Track on Datasets and Benchmarks.

# 1  Introduction

Today, learning and perfecting a new physical skill requires a significant amount of time, practice, and often guidance from an expert coach/professional in that domain. Unfortunately, personalized coaching remains inaccessible to the majority due to prohibitively expensive costs and a lack of expert availability. Recent advances in AI have sparked interest in developing virtual AI assistants/-coaches, particularly in the text domain [35, 2, 46]. However, text-based large language models (e.g., ChatGPT) are insufficient for learning new *physical skills* as standard LLMs typically lack a nuanced understanding of physical human activities/skills. In contrast, rapidly improving vision-language models (VLMs) could serve as valuable tools for various physical skill learning applications. In particular, by recording a video of a skill demonstration and feeding it to a VLM, people could receive detailed, actionable feedback similar to that of expert human coaches.

Although recent progress has enabled modern VLMs to achieve impressive general image/video recognition capabilities [20, 46, 29, 3, 27, 9], existing studies reveal that such VLMs still struggle to understand fine-grained human activities [6, 36], particularly activities that require expert-level knowledge [5, 19, 26]. This is primarily due to the inadequate underlying visual representations that 1) do not capture expert-level knowledge needed to generate feedback for physical skill learning, and 2) are unable to recognize subtle details in skilled human actions.

In addition to the limitations of existing VLMs, there is a notable lack of evaluation benchmarks tailored for expert-level understanding of skilled human activities. As shown in Table 1, most existing datasets focus on coarse activity recognition [8, 33, 45, 21, 34, 43, 51], which typically only require scene-level recognition rather than fine-grained understanding. While several fine-grained video recognition datasets exist [15, 42, 25, 6], the majority of them are aimed at generic (*e.g.*, putting something into something) rather than expert-level action understanding of skilled human activities (*e.g.*, keeping balance during a dance spin, playing the correct rhythm in a piano piece). Beyond action recognition, a number of video-based skill assessment benchmarks have been recently developed [1, 12, 50, 4, 36, 13]. Most of such skill assessment datasets focus on predicting scalar/categorical performance scores to quantify execution quality. Additionally, these datasets

| Dataset | Expert-level Knowledge | Free-form Language Annotations | MCQ Evaluation |
| --- | :---: | :---: | :---: |
| *Coarse Action Recognition Datasets* | | | |
| Kinetics-700 [8] | ✗ | ✗ | ✗ |
| HowTo100M [33] | ✗ | ✗ | ✗ |
| UCF101 [45] | ✗ | ✗ | ✗ |
| HMDB [21] | ✗ | ✗ | ✗ |
| Moments in Time [34] | ✗ | ✗ | ✗ |
| Hollywood [43] | ✗ | ✓ | ✗ |
| ActivityNet-QA [51] | ✗ | ✗ | ✓ |
| *Fine-grained Action Recognition Datasets* | | | |
| Something-SomethingV2 [15] | ✗ | ✗ | ✗ |
| FineGym [42] | ✗ | ✗ | ✗ |
| Multisports [25] | ✗ | ✗ | ✗ |
| TemporalBench [6] | ✗ | ✓ | ✓ |
| *Video-Based Skill Assessment Datasets* | | | |
| JIGSAWS [1] | ✓ | ✗ | ✗ |
| Best [12] | ✓ | ✗ | ✗ |
| FineDiving [50] | ✓ | ✗ | ✗ |
| FP-Basket [4] | ✓ | ✗ | ✗ |
| BASKET [36] | ✓ | ✗ | ✗ |
| Aifit [13] | ✓ | ✓ | ✗ |
| *Skilled Activity Video-Language Datasets* | | | |
| VidDiffBench [5] | ✓ | ✓ | ✗ |
| EgoExo-Fitness [26] | ✓ | ✓ | ✗ |
| EgoExolearn [19] | ✓ | ✓ | ✗ |
| Ego-Exo4D [16] | ✓ | ✓ | ✗ |
| **EXACT (Ours)** | ✓ | ✓ | ✓ |

Table 1: Compared to previous action recognition and skill assessment datasets, our proposed EXACT benchmark uniquely combines expert-level, free-form language annotations and a multiple-choice question (MCQ) evaluation format, making it an excellent resource for evaluating modern video-language models at expert-level understanding of skilled human activities.

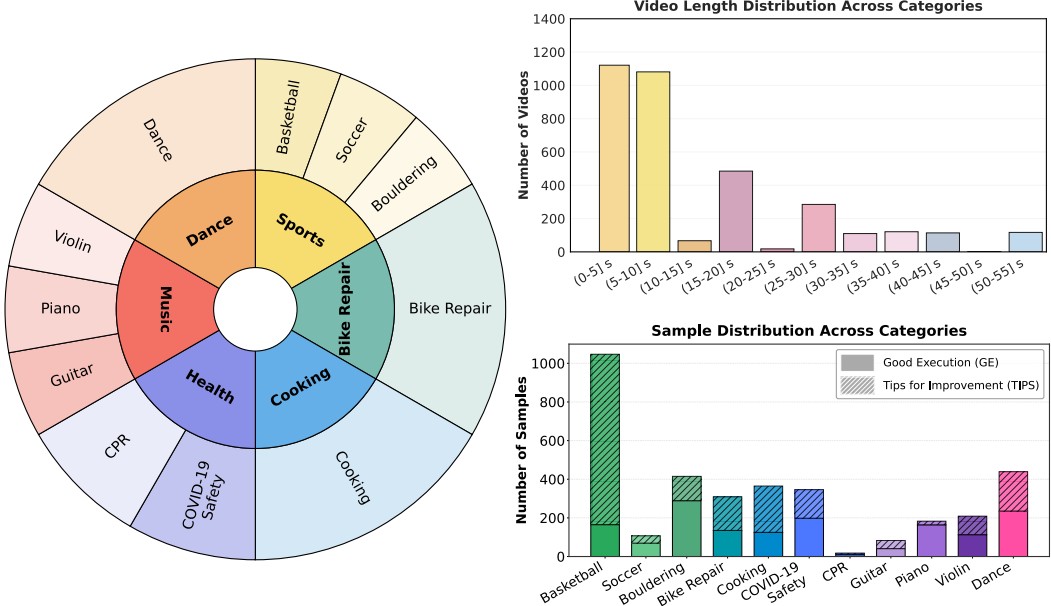

Figure 2: **Left:** Our proposed EXACT benchmark contains 11 skilled activity types spanning 6 broader physical domains: Sports, Music, Dance, Health, Cooking, and Bike Repair. **Top Right:** Distribution of video lengths across the dataset, showing that most clips fall within the 0–10 second range. **Bottom Right:** Sample distribution per activity, categorized by the expert feedback type: Good Execution (GE) and Tips for Improvement (TIPS).

often lack open-ended language annotations, which can provide a rich and intuitive medium to convey skill-specific feedback [5] and can capture subtle errors, temporal nuances, and intent—details that scalar or categorical labels often miss. Finally, a few recent skilled activity video-language datasets [5, 26, 19, 16] use experts to obtain free-form language descriptions akin to verbal coach feedback but do not provide rigorous evaluation benchmarks or tasks, making it difficult to assess how well modern video models can understand nuanced physical human skills.

To address these issues, we introduce EXACT, a video-language benchmark consisting of 3,521 video–question–answer (VQA) pairs designed to evaluate expert-level understanding of skilled physical human actions (sample questions shown in Figure 1). EXACT covers 11 activities across 6 diverse physical domains: Sports (Basketball, Soccer, Bouldering), Bike Repair, Cooking, Health (COVID-19 Safety, CPR), Music (Guitar, Piano, Violin), and Dance, as shown in Figure 2. To construct EXACT, we build on the fine-grained expert commentaries derived from the Ego-Exo4D dataset [16]. The original Ego-Exo4D expert commentaries have several crucial limitations: 1) the original expert commentaries are unstructured and lengthy, 2) they contain automatic speech recognition (ASR) errors, and 3) they include redundant or irrelevant content that is not directly tied to the observed actions. Furthermore, evaluating the quality of open-ended captions/descriptions in the original form of Ego-Exo4D expert commentaries is inherently difficult using standard language metrics such as CIDEr [47], BLEU [37], or ROUGE [28], which do not accurately reflect the correctness or relevance of the instructional feedback in the context of skilled physical activities. To address these challenges, we first apply a structured annotation pipeline that rewrites the original commentaries into concise, self-contained feedback commentaries. Then, our proposed EXACT benchmark evaluates expert action understanding as a multiple-choice question-answering task, which uses a well-defined metric of question-answering accuracy. This multiple-choice question-answering format eliminates the ambiguity and reproducibility issues associated with open-ended evaluation by providing clearly defined answers and controllable question difficulty through the design of distractor options.

We evaluate several state-of-the-art VLMs, including Gemini 2.5 Pro [11], GPT-4.1 [20], GPT-4o [20], Gemini 1.5 Pro [46], LLaVA-Video [55], LLaVA-OneVision [22], Qwen2.5-VL [3], VideoL-LaMA [53], InternVL2.5 [9], and PerceptionLM [10] on our new EXACT benchmark. Our results reveal that compared to human experts, most modern VLMs achieve poor results on EXACT. In particular, Gemini 2.5 Pro, the best-performing model in our experiments, achieves 55.35% accuracy. In comparison, non-expert humans achieve 61.86% accuracy, while domain experts achieve 82.02% accuracy. This substantial gap highlights the limitations of current VLMs in expert-level understanding of physical human skills. We hope that EXACT will serve as a rigorous evaluation benchmark for

measuring expert-level understanding of skilled human actions, thus laying the foundation for AI systems that support enhanced human skill learning.

## 2   Related Work

**Vision-Language Models (VLMs).** Recent advances in Vision-Language Models (VLMs) have demonstrated impressive capabilities in understanding visual content. Models such as Gemini 2.5 Pro [11], GPT-4.1 [20], GPT-4o [20], Gemini 1.5 [46], LLaVA-OneVision [22], Qwen2.5-VL [3], VideoLLaMA [53], and InternVL2.5 [9] have achieved strong performance in tasks such as action recognition, video captioning, and visual question answering. Although these models show remarkable generalization, their outputs are often limited to high-level descriptions of the image/video content, rather than a detailed, fine-grained understanding of physical human actions and skills. More recent video-centric variants, such as LLaVA-Video [55] and VideoLLaMA [53], attempt to extend static image capabilities to temporal inputs. However, these models still struggle to understand the fine-grained nuances of complex physical human activities, which are essential for human skill understanding. Specifically, most existing VLMs cannot assess how well a task is performed, nor articulate the strengths of the execution and the areas requiring improvement [5, 6, 36]. To address this gap, our proposed EXACT provides a rigorous benchmark to evaluate expert-level understanding of physical human skills.

**Multimodal and VQA Benchmarks.** Diverse multimodal benchmarks have emerged to evaluate VLM performance, particularly in the video domain. ActivityNet-QA [51] assesses the temporal reasoning of activities within longer videos via question-answering. NExT-QA [49] focuses on compositional temporal reasoning. STAR [48] emphasizes situated reasoning. Benchmarks such as PerceptionTest [41] probe the perceptual abilities of VLMs in various modalities. TemporalBench [6] targets fine-grained temporal understanding. MV-Bench [24] assesses temporal comprehension across 20 challenging tasks. EgoSchema [31] focuses on egocentric human actions. MMWorld [17] aims to evaluate embodied agents in simulated environments. MLVU [57], Video-MMMU [18], MMVU [56], Video-MMLU [44], and Video-MME [14] aim to evaluate modern MLLMs for complex video question answering tasks. Finally, SEED-Bench [23] and MMBench [30] evaluate various multimodal abilities of MLLMs. However, most existing benchmarks focus on factual recall (e.g., "What color is the car?"), event-level understanding (e.g., "What action is being performed?"), or basic temporal reasoning (e.g., "What happened before this?"), and are not designed to capture the subtle nuances of skilled human actions. In contrast, our newly proposed EXACT benchmark focuses explicitly on expert-level analysis of physical human skills.

**Video-based Skill Assessment Benchmarks.** Several recent works have focused on developing methods to assess human skills from video. Benchmarks such as MITDive [40], UNLV-Dive [39], MTL-AQA [38], and FineDiving [50] provide temporally segmented videos with action labels or action quality scoring for diving. FP-Basket [4] and BASKET [36]) focus on basketball, while LOGO [54]) provides human judgment scores for artistic swimming. Similar efforts also exist in other sports, including figure skating [40] and golf [32]. Furthermore, datasets such as JIGSAWS [1], BEST [12], and EgoExoLearn [19] focus on scenarios beyond sports, such as surgical tasks and daily activities. Most recently, the Ego-Exo4D [16] dataset introduces large-scale egocentric and exocentric video of skilled human activities. Ego-Exo4D includes spoken expert commentaries, offering a unique expert-level supervisory signal for understanding human skills. However, these expert commentaries are typically highly unstructured, noisy due to ASR errors, and often contain irrelevant information. Moreover, Ego-Exo4D does not provide a formal evaluation benchmark/task associated with such expert commentaries. In our work, we leverage such unstructured expert commentaries and construct a rigorous, expert-validated, and easy-to-evaluate EXACT benchmark enabling evaluation of expert-level understanding of physical human actions/skills.

## 3   EXACT Benchmark Construction

We construct EXACT using a four-stage pipeline. In stage I, we pre-process raw expert commentaries using GPT-4o, correcting ASR errors and segmenting them into concise, self-contained feedback commentaries. In Stage II, we construct multiple-choice QA pairs, each consisting of one correct commentary and four distractors. In Stage III, we remove low-quality or biased samples through length filtering and blind-LLMs. Finally, in Stage IV, domain experts review each QA pair to ensure

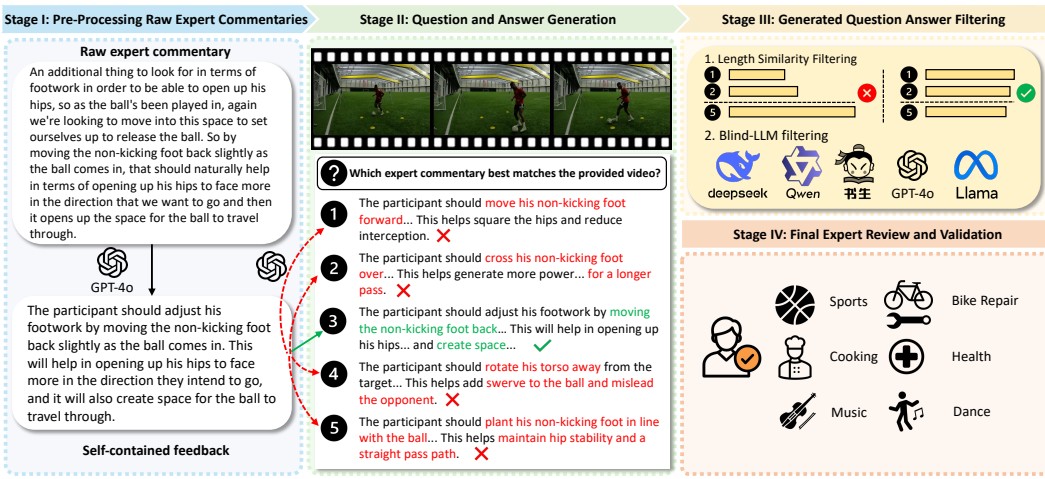

Figure 3: Overview of our benchmark construction pipeline. In stage I, we pre-process raw expert commentaries using GPT-4o, correcting errors and segmenting them into concise, self-contained feedback commentaries. In Stage II, we construct multiple-choice QA pairs, each consisting of one correct expert commentary and four carefully generated distractors. The four red arrows indicate the LLM-generated distractors, while the green arrow represents the correct expert commentary. In Stage III, we filter out low-quality or biased samples using length-based heuristics and blind-LLMs. Finally, in Stage IV, domain experts review all QA pairs to ensure visual grounding and linguistic accuracy.

visual grounding and linguistic accuracy. Our benchmark construction pipeline is illustrated in Figure 3.

## 3.1 Stage I: Pre-Processing Raw Expert Commentaries

The transcribed commentaries from Ego-Exo4D are often lengthy, noisy, and unstructured. They may also include automatic speech recognition (ASR) errors, redundant phrases, and off-topic remarks, as illustrated in the upper-left example of Figure 3. In the first stage of our benchmark construction pipeline, we use an LLM to refine raw expert commentaries from Ego-Exo4D into a more compact and less noisy format. Specifically, to do this, we prompt GPT-4o to 1) correct transcription mistakes, 2) remove irrelevant or repetitive content, and 3) segment the cleaned text into concise, self-contained feedback commentaries. Each commentary is then also assigned to one of the two categories: *good execution* or *tips for improvement*, reflecting the two main ways in which experts deliver their feedback, i.e., by affirming what was done well and/or suggesting what can be improved. We then use such compact and structured commentaries to construct multiple-choice question-answer pairs as described next. The complete prompt templates are provided in the supplementary material.

## 3.2 Stage II: Question and Answer Generation

After pre-processing raw expert commentaries in Stage I, we proceed with multiple-choice QA pair construction. Specifically, we treat the structured commentary from Stage I as a positive commentary and ask the LLM (e.g., GPT-4o) to generate four distractor commentaries that require fine-grained expert-level understanding to distinguish the correct answer from the incorrect ones. This leads to a 5-way multiple-choice QA setup. To create negative commentaries, we use the following strategies:

For expert commentaries assigned to the *good execution* category (see Subsection 3.1), we apply two strategies to generate negative commentaries:

- **Action replacement:** A key action in the original commentary is substituted with a plausible but incorrect alternative (*e.g.*, replacing "... *performs a three-point shot*" with "... *performs a layup*").

- **Absent-action insertion:** A new event or action is inserted that was never mentioned or shown (*e.g.*, adding "... *tightens the brake lever* ..." to "... *keeps the bike steady on a repair stand* ...").

For commentaries in the *tips for improvement* category, we apply four alternative strategies:

- **Action misinterpretation:** Misinterpreting the mistake in an execution. *Example:* Replacing "... *elbow is too low ...*" with "... *grip is too tight ...*".
- **Incorrect technical reasoning:** Correctly identifying a flaw but providing an implausible or technically inaccurate explanation. *Example:* "... *doesn't bend knees ... reduces jump height*" vs. "... *doesn't bend knees ... prevents ball spin*" (i.e., knee movement affects jumping, not ball spin).
- **False cause–effect relationship:** Introducing a misleading causal link between the error and an unrelated factor. *Example:* "... *feet are misaligned, leading to an off-balance shot*" vs. "... *doesn't tuck in jersey, leading to an off-balance shot*".
- **Ineffective suggestion:** Proposing a correction to the execution that does not address the problem. *Example:* "... *doesn't keep eyes on the ball ...*" vs. "... *should stand closer to the baseline*".

Albeit simple in format, our questions effectively probe multi-faceted reasoning abilities—spanning temporal, causal, spatial, and domain-specific understanding—beyond mere visual recognition (see supplementary material S4 for examples).

## 3.3 Stage III: Generated Question Answer Filtering

After constructing initial QA samples (Subsection 3.2), we apply several additional filtering steps to ensure high-quality and unbiased samples. Prior studies [7, 52] have shown that large language models (LLMs) can often exploit subtle statistical or stylistic patterns in multiple-choice answers to correctly identify the ground-truth option without relying on visual input. Such language-driven shortcuts, often referred to as language-related bias, pose a serious threat to the integrity of robust evaluation. To mitigate such biases, we adopt two filtering strategies:

1) **Length similarity:** Similar to human test-takers, LLMs may exploit surface-level cues such as answer length to eliminate implausible distractors. To mitigate this bias, we enforce a length similarity constraint: each distractor must be between 80% and 120% the length of the positive (i.e., correct) expert commentary, and the absolute word count difference must not exceed 8 words. QA samples that violate this constraint are excluded from the benchmark. Note that the 80–120% length range and 8-word absolute difference threshold were determined empirically through iterative pilot studies with human annotators.

2) **Blind-LLM filtering:** Inspired by findings from TemporalBench [6], we observe that LLMs can sometimes identify the correct answer by detecting shared linguistic patterns, particularly when distractors are lightly edited variants of the ground truth. To avoid such language-driven biases, we present each QA sample consisting solely of the five textual options without any video to 5 state-of-the-art LLMs (GPT-4o and DeepSeek-VL-R1 (proprietary), Qwen2.5-72B, LLaMA3.3-70B, and InternLM2.5-20B (open-source, all ranked highly on multiple LLM leaderboards)). Each model is prompted with the question: *"Which expert commentary best matches the provided video?"* If more than 20% of the models (i.e., exceeding the random chance of selecting the correct answer in a 5-way setup) select the positive expert commentary, we consider the sample susceptible to language-only bias and remove it from the dataset.

## 3.4 Stage IV: Final Expert Review and Validation

In the final stage, we conduct a two-phase manual review to ensure that each QA question is clearly formulated and can be reliably answered through expert-level video analysis. Unlike the previous stages, which rely solely on textual inputs, this phase gives domain experts full access to the video alongside the five answer options. This setup allows for a comprehensive multimodal evaluation of answer correctness and distractor plausibility.

First, for each QA sample, domain experts meticulously review the corresponding video clip and all five answer options. Their task is to select the option most consistent with the visual information presented in the video. After submitting their answer, the ground-truth label is revealed. The expert is then asked to verify three criteria: (1) whether the visual content of the video clip supports the correctness of the positive (i.e., correct) expert commentary; (2) whether any of the distractor commentaries also describe actions that are visible or valid within the video; and (3) whether all five candidate answer options are free from grammatical, logical, or instructional flaws. The sample will be removed if any of these criteria are not met. The second criterion is especially important because if a distractor candidate also describes something that appears in the video, the question becomes

ill-posed due to multiple correct answers. For example, suppose that the positive expert commentary states, "*The participant plants their non-kicking foot beside the ball*," while a distractor commentary states, "*The participant plants their foot behind the ball*." In this case, both candidate commentaries are visually observable, which means that the expert cannot answer it using a single answer. We remove all such samples to avoid ambiguity.

This verification process is conducted by 16 experts, each assigned to one of the 11 activity categories based on their area of expertise. Each sample is reviewed by at least one qualified expert to ensure consistency and domain relevance. For domains reviewed by two experts, we observed that experts reached agreement in over 90% of cases—demonstrating strong consistency in their judgments. A screenshot of the annotation interface, along with additional implementation details, is provided in the supplementary material.

# 4   Experimental Setup

**Evaluation Metrics.** We use standard question-answering accuracy as the primary evaluation metric. Our benchmark includes a total of 3,521 QA samples spanning 11 fine-grained physical skilled activities. For each activity, we compute the accuracy as the percentage of questions for which the model selects the correct answer. To summarize performance across the dataset, we report the average accuracy across all 11 activities. We additionally report per-domain accuracy (Sports, Bike Repair, Cooking, Health, Music, and Dance) to highlight domain-specific generalization.

**Baseline Models.** To thoroughly assess the challenges posed by EXACT, we evaluate various state-of-the-art Video-Language Models (VLMs), including proprietary models: Gemini 2.5 Pro [11], GPT-4.1 [20], GPT-4o [20], Gemini 1.5 Pro [46], and open-source models: LLaVA-Video [55], LLaVA-OneVision [22], Qwen2.5-VL [3], VideoLLaMA [53], InternVL2.5 [9], and PerceptionLM [10]. These models vary significantly in architecture, training corpus, and modality integration strategies, offering a broad and representative basis to evaluate expert-level feedback capabilities.

**Implementation Details.** All model inferences are conducted using 4 NVIDIA Tesla H100 GPUs, each with 96 GB of memory. For fairness, we adopt uniformly sampling strategy and extract 32 frames per video clip for all models. Each frame is extracted at a resolution of $796 \times 448$ and then resized internally according to the input resolution requirements of each model. Unless otherwise specified, we use the same prompt template and follow the official inference code provided by each model. We also ablate on several key hyperparameters in Subsection 5.4. Additional implementation details, including full prompt formulations, are provided in the supplementary material.

# 5   Experimental Results

## 5.1   Main Results

In Table 2, we report the performance of several state-of-the-art video-language models (VLMs) on our EXACT benchmark. Overall, none of the methods achieve accuracy exceeding 56%. Among the evaluated models, Gemini 2.5 Pro achieves the highest overall accuracy at 55.35%. This model outperforms all others across all six domains. Notably, it achieves over 60% accuracy in the domains of Bike Repair and Health. These domains tend to rely more heavily on procedural knowledge, such as understanding sequences of actions and planning the next steps. In contrast, domains like Sports, Cooking, Music, and Dance involve highly specialized, fine-grained physical movements that require detailed visual perception and precise temporal understanding.

We also observe that proprietary models (e.g., GPT and Gemini) consistently outperform the best-performing open-source alternatives. This performance gap highlights the limitations of current public models in capturing the fine-grained, domain-specific reasoning required for expert-level skill understanding. Additionally, model size appears to be a significant factor in fine-grained video comprehension. Smaller models such as PerceptionLM-8B [10] and VideoLLaMA3-7B [53] achieve only 24.65% and 26.38% accuracy respectively—barely above random chance. In comparison, larger open-source models reach at least 33% accuracy, suggesting that scale contributes to more effective representation and complex activity analysis.

| Model | Overall (%) | Results by Domain (%) | | | | | |
|---|---|---|---|---|---|---|---|
| | | Sports | Bike Repair | Cooking | Health | Music | Dance |
| Random Choice | 20.00 | 20.00 | 20.00 | 20.00 | 20.00 | 20.00 | 20.00 |
| Human Non-Expert | 61.86 | 62.97 | 55.02 | 66.58 | 71.43 | 54.11 | 59.22 |
| Human Expert | 82.02 | 82.09 | 81.23 | 80.27 | 87.09 | 80.21 | 81.55 |
| **Open-source VLMs** | | | | | | | |
| PerceptionLM-8B [10] | 24.65 | 24.22 | 28.16 | 25.75 | 22.53 | 22.95 | 26.42 |
| VideoLLaMA3-7B [53] | 26.38 | 26.64 | 23.30 | 29.32 | 26.65 | 23.79 | 27.79 |
| InternVL2.5-78B [9] | 33.48 | 31.93 | 36.57 | 33.70 | 37.91 | 32.00 | 34.62 |
| LLaVA-OneVision-72B [22] | 35.44 | 33.65 | 43.04 | 33.42 | 35.44 | 30.53 | 43.51 |
| Qwen2.5-VL-72B-Instruct [3] | 35.67 | 35.62 | 37.86 | 33.97 | 36.26 | 32.63 | 38.50 |
| LLaVA-Video-72B [55] | 41.58 | 41.81 | 42.72 | 44.11 | 32.42 | 38.74 | 48.52 |
| **Proprietary VLMs** | | | | | | | |
| Gemini 1.5 Pro [46] | 43.91 | 42.83 | 52.10 | 51.78 | 41.21 | 41.89 | 39.86 |
| GPT-4o [20] | 44.70 | 43.47 | 52.75 | 46.30 | 53.30 | 33.89 | 46.70 |
| GPT-4.1 [20] | 50.89 | 51.37 | 58.90 | 54.25 | 51.10 | 40.84 | 51.48 |
| Gemini 2.5 Pro [11] | **55.35** | **52.58** | **65.05** | **58.36** | **60.71** | **53.05** | **53.98** |

Table 2: EXACT evaluation results (QA accuracy) across six diverse physical domains: Sports (Basketball, Soccer, Bouldering), Bike Repair, Cooking, Health (COVID-19 safety, CPR), Music (Guitar, Piano, Violin), and Dance. The results show a significant gap between the performance of modern Vision-Language Models (VLMs) and human experts, indicating that there is a significant room for improvement for future video-language models.

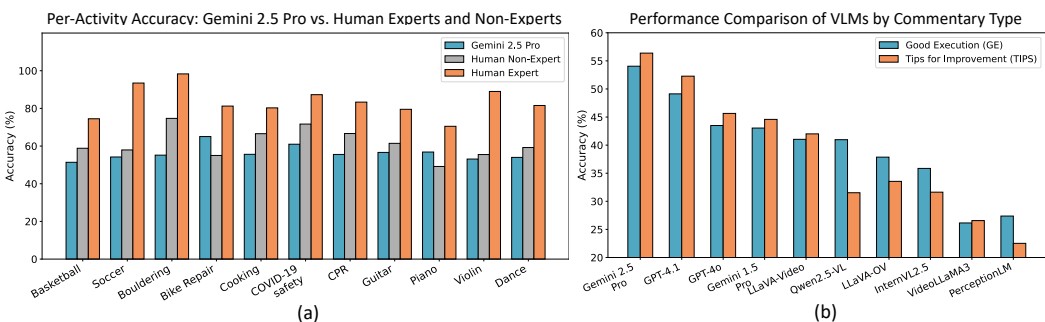

Figure 4: (a) Accuracy of the best-performing model (Gemini 2.5 Pro) compared to human experts and non-experts across domains. While Gemini 2.5 Pro achieves the highest accuracy among VLMs, it still falls short of human performance: experts consistently achieve over 80% accuracy in most domains, with soccer and bouldering reaching up to 95%. (b) Performance of VLMs on Good Execution (GE) and Tips for Improvement (TIPS) commentary categories. Models such as PerceptionLM, InternVL2.5, LLaVA-OneVision, and Qwen2.5-VL perform better on GE, while all proprietary models show stronger performance on TIPS.

## 5.2 Human Performance Analysis

To quantify the gap between human and model performance, we conduct a human evaluation involving two groups: **experts**, comprising trained coaches and professionals with significant domain expertise (i.e., $> 10\ years$), and **non-experts**, individuals without professional expertise in a given domain. As shown in Figure 4 (a), experts achieve consistently high accuracy—often exceeding 80% across most domains, with soccer and bouldering reaching around 95%. Additionally, we observe that while non-experts perform considerably worse (60-70% accuracy), they still surpass the best-performing VLM model (i.e., Gemini 2.5 Pro). These results highlight a substantial gap between human and VLM capabilities, emphasizing the challenges of EXACT and the limitations of current VLMs.

| Num. Frames | Acc. (%) |
|---|---|
| 8 | 39.22 |
| 16 | 39.24 |
| **32** | **41.58** |
| 64 | 39.73 |

| Model Size | Acc. (%) |
|---|---|
| 7B | 27.07 |
| **72B** | **41.58** |

| Prompting | Acc. (%) |
|---|---|
| w/o Time Info | 40.04 |
| w/ Time Info | **41.58** |

(a) **Number of Frames:** Using 32 input video frames leads to the best accuracy.

(b) **Impact of LLM size:** Using larger LLMs leads to significantly better performance.

(c) **Prompting strategies:** Incorporating time information into the prompt improves accuracy.

Table 3: Ablation study on different design choices: number of input frames, impact of LLM size, different prompt strategies.

## 5.3 Performance by Expert Commentary Type

Figure 4 (b) shows the performance of various VLMs across two expert commentary categories: Good Execution (GE) and Tips for Improvement (TIPS). Models such as PerceptionLM [10], InternVL2.5 [9], LLaVA-OneVision [22], and Qwen2.5-VL [3] perform better on GE samples. However, they struggle with questions from the TIPS category, which require identifying subtle mistakes in skilled activity executions. In contrast, stronger models such as Gemini 2.5 Pro [11] and GPT-4.1 [20] perform better on questions from the TIPS category, indicating a greater capacity for expert-level understanding of errors/mistakes in physical human activities.

## 5.4 Ablation Studies

In Table 3, we conduct ablation studies on four key design choices: (a) number of input frames, (b) impact of LLM size, (c) different prompt strategies, and (d) spatial video resolution. For (a)–(c), we use LLaVA-Video [55], the strongest-performing open-source baseline. We ablate the effect of spatial video resolution using Qwen2.5-VL, which supports native-resolution inputs.

**Number of Input Frames.** Table 3a shows the effect of varying the number of input frames using a uniform sampling strategy. We observe a consistent improvement in performance as the number of frames increases, with the model achieving the highest accuracy at 32 frames. This suggests that incorporating more temporal context enhances the model's ability to understand skilled human actions. We also observe that increasing the number of frames beyond 32 does not yield further gains, which may be because most VLMs are not optimized to process longer video sequences effectively.

**Impact of LLM Size.** In Table 3b, we use the same model architecture while varying the size of LLM in LLaVA-Video: 7B and 72B. As the size of LLM parameters increases, we observe a consistent improvement in accuracy. This trend highlights the critical role of the language model for capturing fine-grained video-language cues necessary for human skill analysis.

**Prompting Strategies.** In Table 3c, we analyze the impact of different prompting strategies on model performance. We first evaluate the role of including time information (*e.g. "The video is {video_time}s long, and {len(frames)} uniformly sampled frames occur at {frame_time}."* in the prompt. Removing the time information leads to a performance degradation of 1.54%, indicating that the understanding of time plays a meaningful role in the model's analysis of human skill.

**Spatial Video Resolution.** We investigate how different spatial input resolutions affect the performance of Qwen2.5-VL by evaluating three settings: the original resolution of $796 \times 448$ (35.67% accuracy), a $1.5\times$ downsampled version at $531 \times 299$ (37.40% accuracy), and a $2\times$ downsampled version at $398 \times 224$ (35.03% accuracy). Interestingly, the model achieves the highest accuracy at the mid-level resolution of $531 \times 299$. We hypothesize that although higher resolutions provide more visual detail, they may lead to performance degradation due to increased token length and a mismatch with the lower-resolution inputs commonly seen during pretraining.

## 6 Conclusion

We introduce EXACT, a new video-language benchmark designed to evaluate expert-level understanding of skilled human activities across a diverse set of physical and procedural domains. Our new benchmark uses fine-grained, expert-level, language annotations and a multiple-choice evaluation

format to enable a rigorous evaluation of expert-level understanding of physical human skills. Our experiments reveal a significant gap between state-of-the-art VLMs and human experts' performance, indicating a significant room for future improvement in video-language model design. We believe that ExAct will be pivotal in the development and evaluation of video language models capable of skilled human activity understanding.

**Limitations.** Although our benchmark spans multiple domains, it captures only a fraction of real-world activities. Additional tasks from underrepresented or specialized fields such as surgery or mechanical engineering may elicit different behaviors from current models and offer further insights. Moreover, certain domains (e.g., COVID-related tasks) may be time-sensitive or outdated, potentially affecting the relevance of some samples. Finally, while all participants consent to data usage, the inclusion of real-world videos containing identifiable human faces raises potential privacy concerns. The ethical implications surrounding the reuse and dissemination of such visual data warrant careful consideration and responsible handling.

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

# ExAct: A Video-Language Benchmark for Expert Action Analysis

## Supplementary Material

Our supplementary materials include the following sections: Section S1: Full Prompts for Constructing ExAct, Section S2: Annotation Interface Details, Section S3: Qualitative Results, Section S4: Evaluated Reasoning Dimensions in ExAct, Section S5: Sample statistics after each filtering stage.

## S1    Full Prompts for Constructing ExAct

We provide the detailed prompts used for (1) pre-processing raw expert commentaries, (2) generating negative/distractor commentaries for ExAct, and (3) formatting inputs for VLMs.

**Prompt for Pre-Processing Raw Expert Commentaries.** We present the prompts used for preprocessing raw expert commentaries into Good Execution and Tips for Improvement in Figure 5 and Figure 7, respectively. Corresponding examples of Good Execution and Tips for Improvement are shown in Figure 6 and Figure 8.

> You are given a raw, transcribed expert commentary from a dataset describing a participant performing an {activity}. These transcriptions may contain automatic speech recognition (ASR) errors, redundant phrases, off-topic remarks, or unstructured language.
> Your primary goal is to identify the good executions mentioned in the commentary and express them in a clean, concise, and coherent manner. Specifically:
>   1. Correct any transcription or grammatical errors.
>   2. Remove irrelevant, repetitive, or filler content.
>   3. Your goal is to determine the good executions mentioned in the commentary and write them in a coherent, concise manner.
>   4. If there are no good executions, respond with: "The expert mentions no good executions."

Figure 5: A prompt for pre-processing raw expert commentaries from the good execution (GE) category.

> **Sample 1**
>
> **Input**
> A grand great rhythm and flow as he crosses over. This is a great picture of going off the correct foot, which is his right foot, raising that left knee. He's got his eye and head up, total great body extension. He's got in great spatial awareness of where he is in relation to the basket and the rim. He's not too far under, he's not too far out.
>
> **Output**
> The participant has a great rhythm and flow as he crosses over. He goes off of the correct foot, raises the correct knee, and has his head and eyes up. Overall, great body extension and spatial awareness under the rim.
>
> **Sample 2**
>
> **Input**
> So here the dancer is jumping side to side, almost like a little short gallop. Really nice leg space. I definitely would recommend bending the knees a little bit more. And then just really like, yeah, I think if we bent the knees that would just kind of bring a better level and ease for him to jump side to side. Good job with the arms swaying back and forth on that part and then good eye connection to the audience facing front.
>
> **Output**
> The participant demonstrates nice leg space during the jumps. Additionally, the participant nicely sways their arms and makes good eye contact with the audience.

Figure 6: Samples of converting raw expert commentaries into concise good execution (GE) commentaries.

You are given a raw, transcribed expert commentary from a dataset describing a participant performing an {activity}. These transcriptions may contain automatic speech recognition (ASR) errors, redundant phrases, off-topic remarks, or unstructured language.

Your primary goal is to identify the tips for improvement mentioned in the commentary and express them in a clean, concise, and coherent manner. Specifically:

1. Correct any transcription or grammatical errors.

2. Remove irrelevant, repetitive, or filler content.

3. Your goal is to determine the tips for improvement mentioned in the commentary and write them in a coherent, concise manner.

4. If there are no tips for improvement, respond with: "The expert mentions no tips for improvement."

Figure 7: A prompt for pre-processing raw expert commentaries from the tips for improvement (TIPS) category.

**Sample 1**

**Input**
This is another example again of the person taking too heavy of a touch and then overextending his leg to be able to wrap around the ball. So actually you see now how he's contacting the ball. Instead of kind of catching it, he's actually having to almost like, his foot placement is much higher on the center of the ball here. He's actually coming like downward with the force in this direction when he's actually contacting it, which could actually potentially make the ball jump up when he hits it.

**Output**
The participant should try to take a lighter touch and focus on not overextending their leg to wrap around the ball. The participant should avoid coming downward when contacting the ball because it could make the ball jump up when they hit it.

**Sample 2**

**Input**
This is a really excellent shift that the player had. If they wanted to be as expressive, they could change the weight as they're going up to come into the fingerboard a little bit more so it comes across as a more expressive slide.

**Output**
The participant should change the weight as they go up to come into the fingerboard so it comes across as a more expressive slide.

Figure 8: Samples of converting raw expert commentaries into concise tips for improvement (TIPS) commentaries.

**Prompt for Negative Commentary Generation.** Here we present the prompts used for negative/distractor commentary generation. For Good Execution (GE) commentaries, we adopt two strategies, as shown in Figure 9. For Tips for Improvement commentaries, we use four strategies, as shown in Figure 10.

> You are a bouldering expert tasked with creating **wrong but plausible commentary** to train a video understanding model. You will be given high-expertise bouldering commentary, and your task is to generate **four wrong comments** based on that expert commentary. These wrong comments should be grounded in visible actions from the video and appear reasonable but must either provide an incorrect justification or directly misinterpret the actions.
>
> **Requirements:**
>
> - All comments must be grounded in observable actions from the video and avoid references to non-visual elements.
> - Match the length, detail, and complexity of the expert's original comments without obvious stylistic differences.
> - The difference between correct and incorrect comments should lie in the reasoning or specific actions mentioned.
> - Do not generate comments that might sometimes be true; ensure the actions are *definitely* incorrect based on expert feedback.
> - Avoid using negative adjective words such as "improper," "bad," "not good," or "not perfect."
> - Ensure the incorrect comments appear plausible, limiting each to 1–2 subtle errors.
> - Vary the type of error across the four generated comments. Keep all comments logical and coherent.
>
> **Some techniques for creating wrong comments:**
>
> - **Action replacement:** A key action in the original commentary is substituted with a plausible but incorrect alternative.
> - **Absent-action insertion:** A new event or action is inserted that was never mentioned or shown.
>
> **Output Format:**
>
> *Good execution:* "The participant demonstrates a good initiation of upward movement, properly preparing their legs to generate momentum upwards."
>
> *Wrong Comments:*
>
> - **Action replacement:** "The participant demonstrates a good initiation of downward movement, correctly preparing their arms to generate momentum downwards."
> - **Action replacement:** "The participant properly prepares their arms to generate momentum sideways, effectively aiding their lateral movement along the boulder."
> - **Absent-action insertion:** "The participant demonstrates a clever strategy by swinging their body to the side before leaping to the next hold, avoiding direct upward movement."
> - **Absent-action insertion:** "The participant initiates a powerful dyno by planting both feet on the foothold and lunging directly for the top of the wall, using agility over controlled movement."
>
> *The high-expertise comment is as follows:*

Figure 9: Prompt for negative/distractor commentary generation for good execution commentaries. We use a bouldering example for illustration.

You are a cooking expert tasked with creating **wrong but plausible commentary** to train a video understanding model. You will be given high-expertise cooking commentary, and your task is to generate **four wrong comments** based on that expert commentary. These wrong comments should be grounded in visible actions from the video and appear reasonable but must either provide an incorrect justification or directly misinterpret the actions.

**Requirements:**

- All comments must be grounded in observable actions from the video and avoid references to non-visual elements.

- Match the length, detail, and complexity of the expert's original comments without obvious stylistic differences.

- The difference between correct and incorrect comments should lie in the reasoning or specific actions mentioned.

- Do not generate comments that might sometimes be true; ensure the actions are *definitely* incorrect based on expert feedback.

- Avoid using negative adjective words such as "improper," "bad," "not good," or "not perfect."

- Ensure the incorrect comments appear plausible, limiting each to 1–2 subtle errors.

- Vary the type of error across the four generated comments. Keep all comments logical and coherent.

**Some techniques for creating wrong comments:**

- **Action misinterpretation:** Misinterpreting the mistake in an execution.

- **Incorrect technical reasoning:** Correctly identifying a flaw but providing an implausible or technically inaccurate explanation.

- **False cause–effect relationship:** Introducing a misleading causal link between the error and an unrelated factor.

- **Ineffective suggestion:** Proposing a correction to the execution that does not address the problem.

**Output Format:**

*Tips for improvement:* "The participant should add herbs, spices, and tea while waiting for the mixture to come to a simmer to improve efficiency and flavor infusion."

*Wrong Comments:*

- **Action misinterpretation:** "The participant should wait until the mixture has finished simmering before adding herbs, spices, and tea, as this prevents any flavors from being cooked out of the ingredients."

- **Incorrect technical reasoning:** "The participant should add herbs, spices, and tea while the mixture is boiling vigorously, as the intense heat heightens the flavor of these ingredients."

- **False cause–effect relationship:** "The participant should add herbs, spices, and tea just after the mixture stops simmering, as this allows the flavors to cool simultaneously with the dish for balanced taste."

- **Ineffective suggestion:** "The participant should blend the herbs, spices, and tea into a smooth paste before adding them to the mixture after it comes to a simmer, ensuring a more uniform flavor throughout."

*The high-expertise comment is as follows:*

Figure 10: Prompt for negative/distractor commentary generation from tips for improvement commentaries. We use a cooking example for illustration.

**VLM Prompt for Processing EXACT.** Here we provide the template for the input prompt to LLaVA-Video. The prompts used for other models are very similar, only with a different video separation token. The default image token is the video separation input for LLaVA-Video. The scenario prompt briefly introduces the activity being performed by the participant (*e.g.*, "The participant is practicing basketball."). We also include time-related instructions, such as the total video duration and the timestamps of uniformly sampled frames in the prompt to guide the VLMs. The prompt presents five candidate answer options labeled **Option 1** to **Option 5**, including one correct answer and four distractors.

```
<DEFAULT_IMAGE_TOKEN>
```
The video is {`video_time`} seconds long, and {`len(frames)`} uniformly sampled frames occur at {`frame_time`}.
{`scenario_prompt`}
Below are different feedback statements about the person's performance in this video:
**Option 1.    Option 2.    Option 3.    Option 4.    Option 5.**
Based on what you observe in the video, which expert commentary best matches the provided video?
**Just respond with the option number (1–5) and nothing else.**

Figure 11: An input prompt to Vision-Language models (VLMs) for processing EXACT.

## S2    Annotation Interface Details

In this section, we provide more details related to the annotation interface used for annotating EXACT. We develop a user-friendly web-based platform tailored for human annotators. A total of 16 experts participated in the annotation process, each with at least ten years of experience in their respective domains. The website is built using `github.io`, with `Formspree` used to collect submission data from annotators. All annotators are compensated at a rate of $50 per hour. The interface begins with an introduction to the project, followed by detailed annotation guidelines. It supports saving progress, allowing annotators to complete their assigned tasks over multiple sessions rather than in a single sitting. Each expert is only assigned samples within their domain of expertise. Upon completing their assigned tasks, annotators submit their responses via the form. Figure 12 and Figure 13 shows our data annotation interface.

**Guidelines for Annotators.** Please follow the instructions below when annotating:

1. Carefully watch the video clip.
2. Read all five options and select the one you believe is correct.
3. Click the "Confirm Selection" button to submit your answer.
4. After submitting your selection, the ground-truth answer will be revealed. Please then evaluate the sample based on the following criteria:
   (a) Does the video clearly support the action described in the ground-truth commentary?
   (b) Do any of the other options also appear valid based on the video?
   (c) Are there any language issues, such as grammatical errors or illogical phrasing, in any of the options?
5. Click "Continue" to move on to the next sample.

# Cooking Technique Evaluation

Please review each cooking video and select the option you believe is most accurate

ⓘ Your progress is automatically saved as you go

**Instructions:**

1. Watch the video clip carefully
2. Read all options and select the one you believe is correct
3. Click the "Confirm Selection" button to submit your answer
4. After your selection, the ground truth will be revealed. Please assess the sample quality by verifying:
   - Whether the video clearly shows the action described in the ground truth
   - Whether any other options (besides the ground truth) are also supported by the video
   - If there are any language issues (grammatically incorrect or illogical options)
5. Click "Continue" to proceed to the next question

Question 1 of 239                                               Progress saved

Figure 12: User instructions for the annotation platform interface.

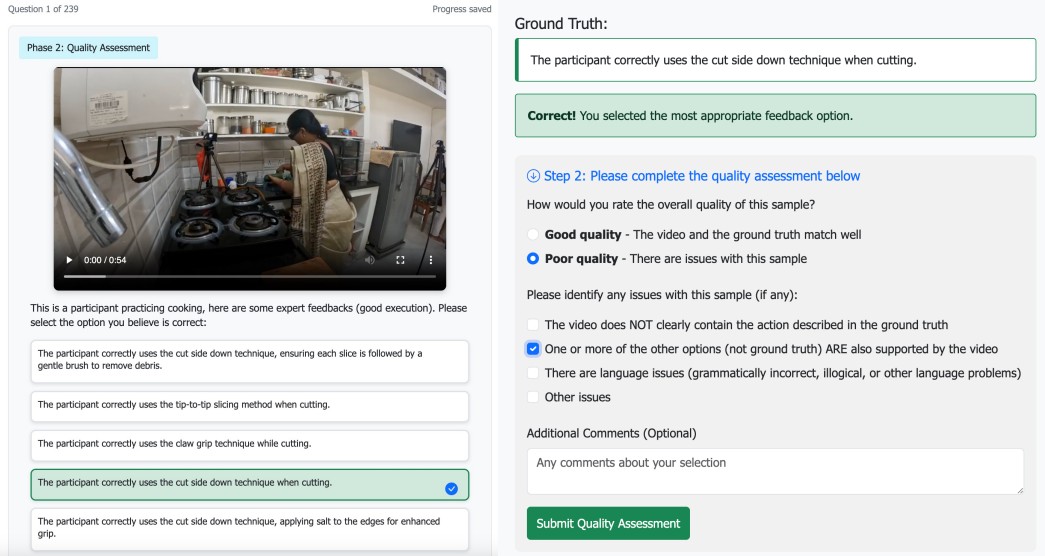

Figure 13: Two-phase manual review interface on the annotation website.

## S3   Qualitative Results

In this section, we present four QA examples (Figures 14–17) that range from easy to difficult.

**Sample 1** (Figure 14) represents a relatively easy example. All models, as well as both human experts and non-experts, correctly identify the correct answer.

**Sample 2** (Figure 15) presents a moderately challenging sample. Only human experts and some of the best models identify the correct answer.

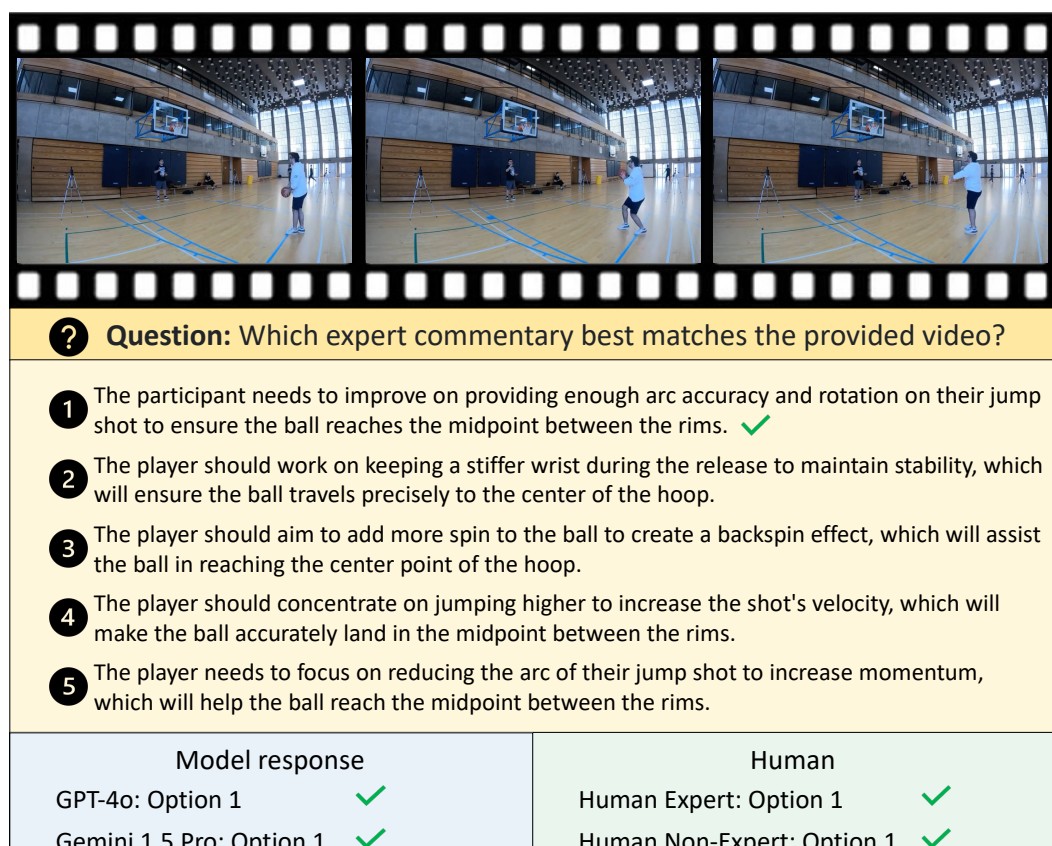

Figure 14: Sample 1 (Basketball): All models, as well as both human experts and non-experts, select the correct answer.

**Sample 3** (Figure 16) presents a more difficult case. Only human experts and GPT-4o select the correct answer.

**Sample 4** (Figure 17) illustrates a particularly difficult case. None of the models are able to select the correct answer. Only human experts succeed. This highlights a significant gap between current VLM capabilities and expert-level understanding, especially for tasks that require nuanced, domain-specific reasoning.

These examples collectively highlight the limitations of current VLMs in expert-level understanding of physical human skills.

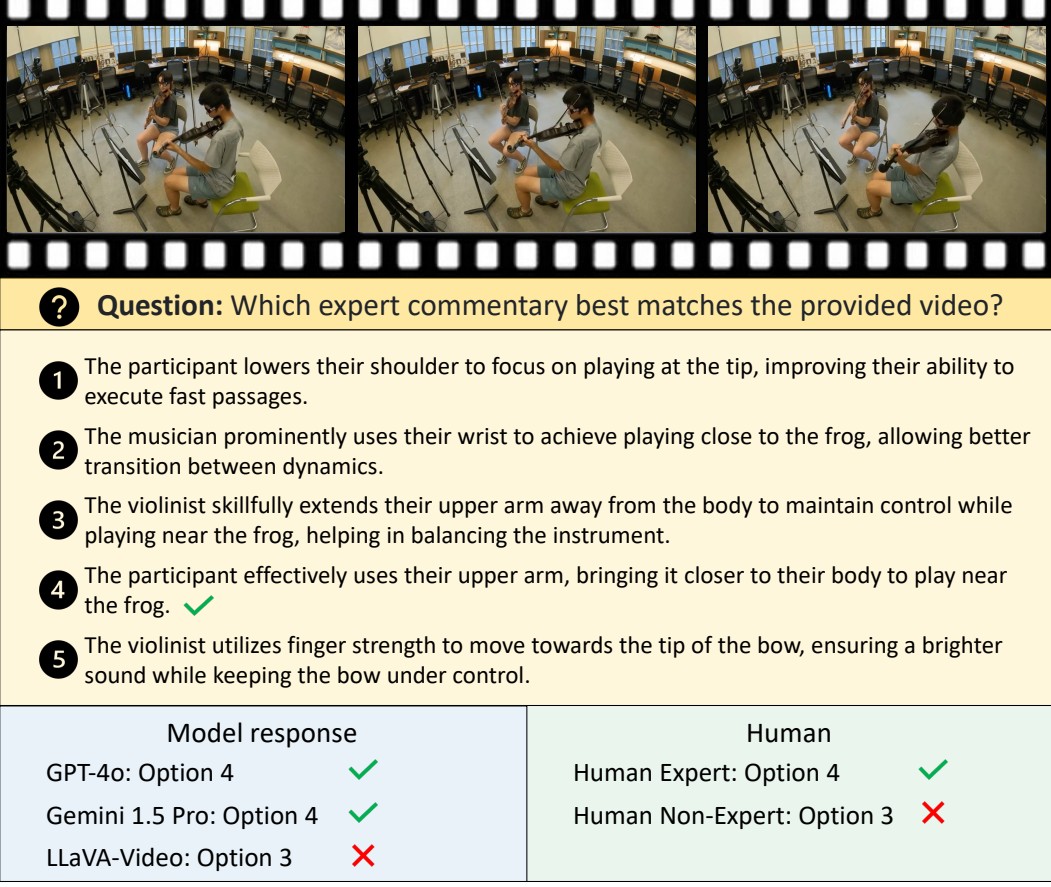

Figure 15: Sample 2 (Violin): Some models and human experts select the correct answer.

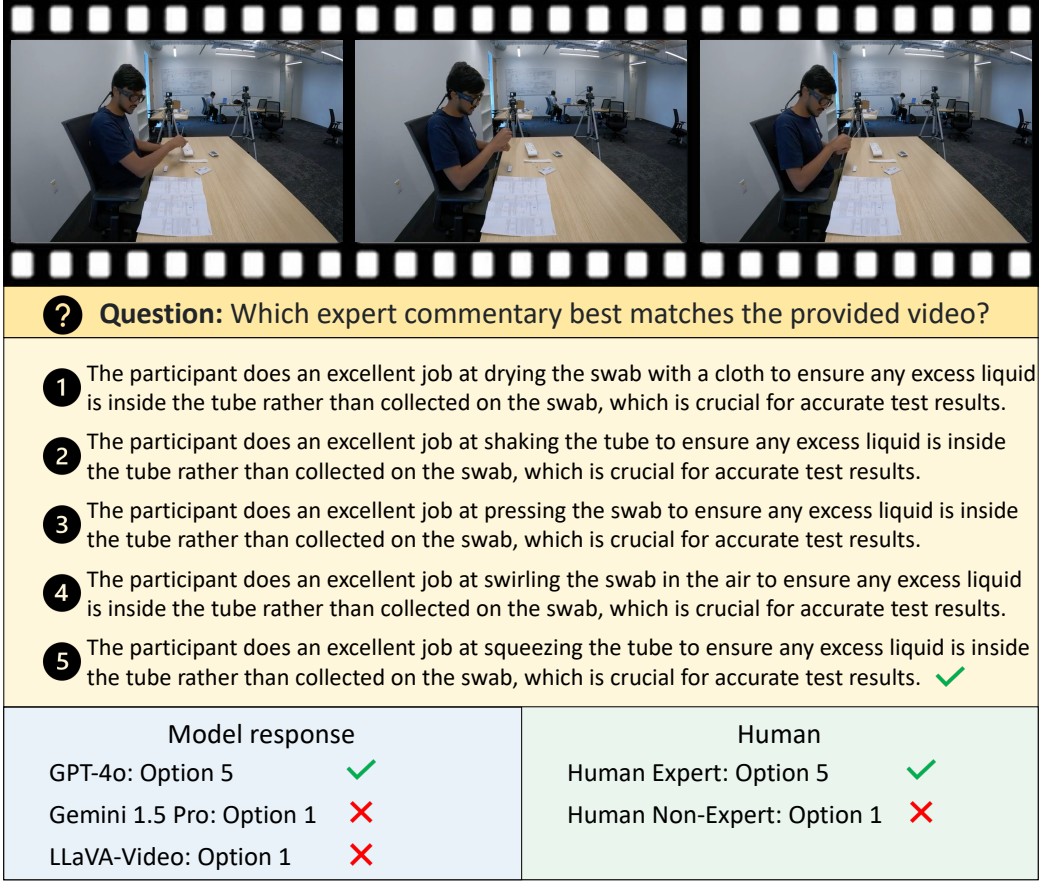

Figure 16: Sample 3 (COVID-19 Safety): Only GPT-4o and human experts select the correct answer.

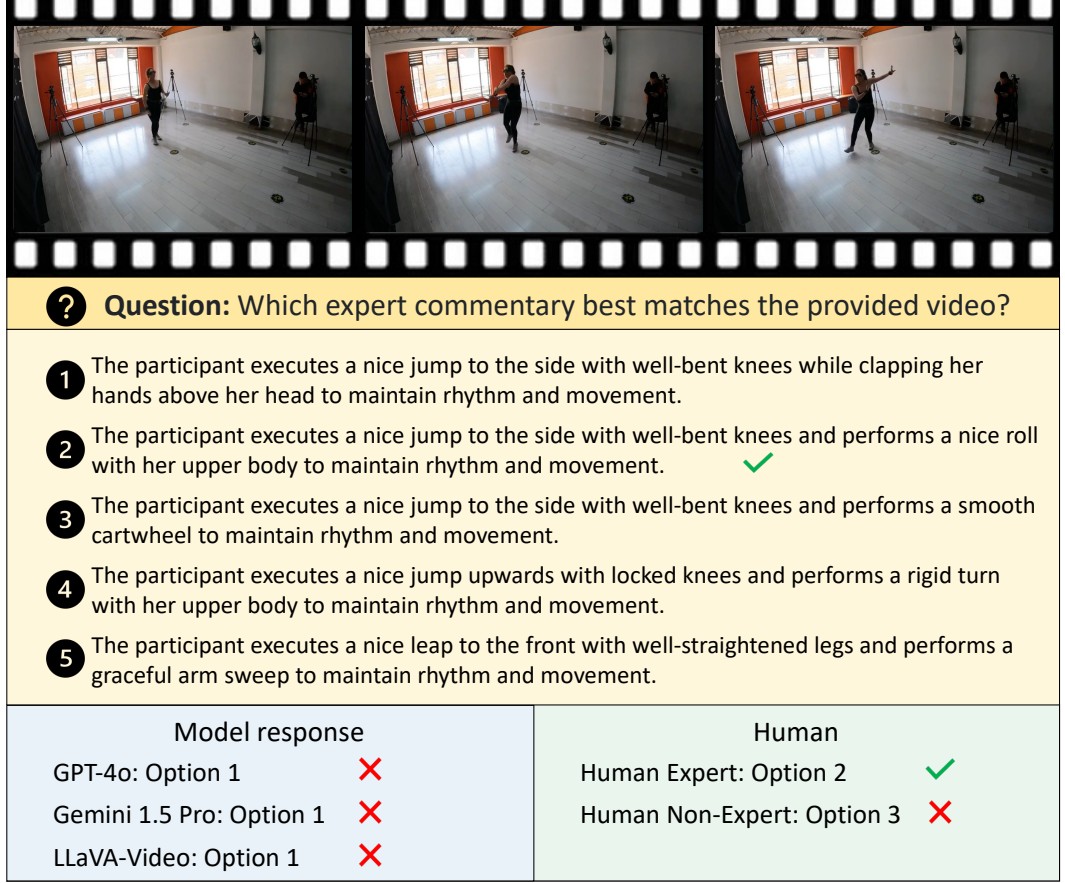

Figure 17: Sample 4 (Dance): None of the models select the correct answer. Only human experts identify the correct response.

## S4 Evaluated Capabilities in EXACT

This section details the reasoning capabilities implicitly evaluated by our QA framework. Although each question follows a simple multiple-choice format, correctly answering them requires diverse capabilities that extend beyond visual recognition. The evaluated reasoning types include:

- **Fine-grained recognition:** Detecting subtle differences in body positioning, movement patterns, and technique execution (e.g., "plays with a very tight wrist" vs. "uses a hybrid picking technique").

- **Temporal reasoning:** Understanding timing precision, action phases, and temporal dependencies (e.g., "releases the ball too early" vs. "too late after the peak jump").

- **Causal understanding:** Linking technique flaws to performance outcomes (e.g., "doesn't bend knees...reduces jump height" vs. "doesn't bend knees...prevents ball spin").

- **Procedural reasoning:** Evaluating adherence to task sequences (e.g., "checks pulse before calling for help" vs. "delays calling for help and requesting an AED").

- **Spatial & geometric reasoning:** Understanding angles, trajectories, and spatial relationships (e.g., "flattens the arc on a left-handed layup" vs. "lowers the entry angle to avoid rotation").

- **Domain-specific expertise:** Applying professional knowledge across sports, music, healthcare, cooking, bike repair, and dance (e.g., "maintains CPR rate at 100–120 bpm" vs. "120–140 bpm").

- **Physical understanding:** Reasoning about biomechanics, force generation, and energy transfer (e.g., "angles knees inward for power" vs. "opens stance toward the rim").

## S5 Sample statistics after each filtering stage

Starting from over 400k expert commentaries in Ego-Exo4D, Stage I reduced the number of samples to 108k ($\approx$ –73%). Stage III's length similarity filtering further reduced it from 108k to 60k ($\approx$ – 44%). The blind-LLM filtering further shrank the sample size from 60k to 4.5k ($\approx$ –92.5%). Finally, Stage IV expert verification led to 3.5k samples ($\approx$ –22% of the 4.5k), leaving about 0.9% of the original samples. These statistics demonstrate that Stage III is critical as it filters low-quality or easily "cheatable" samples before sending them to the experts for final verification.

