# OpenReview forum: "ExAct: A Video-Language Benchmark for Expert Action Analysis"
_NeurIPS.cc/2025/Datasets_and_Benchmarks_Track — NeurIPS 2025 Datasets and Benchmarks Track poster_

### Official Review · Reviewer_PK3K · 2025-07-01

**Rating:** 5
**Confidence:** 5

**Summary:**

This paper introduces EXACT, a novel video-language benchmark specifically designed to assess AI models' expert-level understanding of skilled physical human activities. EXACT evalutes complex human skills comprehension by covering 11 diverse activities across 6 distinct physical domains. The authors conducted extensive empirical evaluations on several state-of-the-art VLMs. The results highlight the current limitations of these models in achieving true expert-level comprehension.

**Dataset Code Accessibility:**

Yes

**Dataset Code Comments:**

The dataset and code links are provided.

**Ethical Considerations:**

No, there are no or only very minor ethics concerns

**Final Justification:**

My concerns were well addressed.

**Limitations Weaknesses:**

1. How many participants were involved in the human evaluation? Was the participant pool sufficiently diverse?
2. The paper lacks detailed statistics about the dataset, such as video resolution, frame rate, and the token length distribution of the text inputs.
3. Were the human evaluators compensated for their participation?
4. Although the benchmark includes more categories than previous work, it still shows limited diversity within each category. The authors could consider expanding this aspect in future work.
5. For categories like dance, music, and sports, are the evaluation questions fine-grained enough to capture expert-level or competition-level detail?

**Strengths Contributions:**

1. A novel video-language benchmark is proposed.
2. Extensive evalutions are conducted on multiple video-language models.
3. The paper is well-writen and structured.
4. The pipeline of dataset construction is clear.

---

> ### Author Rebuttal · Authors · 2025-07-30
>
> **Human annotation details.**
>
> To clarify, only the 16 domain experts and professional coaches were involved in the annotation phase; non-experts did not participate in this stage. In the evaluation phase, human performance was measured using two distinct groups: domain experts (the same individuals from the annotation team) and 12 university students from diverse backgrounds as non-expert evaluators. For domain experts, evaluation and annotation were performed in a unified workflow. As described in the supplementary material (pages 25–26), experts were first asked to complete the multiple-choice questions without seeing the correct answers. After submitting their responses, the correct answer was revealed, and they were then asked to annotate whether the QA sample was valid, well-posed, and challenging. This procedure ensured that expert evaluations were unbiased while simultaneously enabling high-quality annotation. All human evaluators were compensated at $50 per hour, with a performance-based incentive doubling the compensation for those achieving an accuracy rate of over 90%. This incentive was designed to encourage careful and thorough assessment. We will revise the paper to clearly reflect these distinctions.
>
> **Detailed dataset statistics.**
>
> As discussed in Lines 50-51, our benchmark is built on top of the Ego-Exo4D dataset. The video specifications are as follows: resolution of 796 × 448 pixels and a frame rate of 30 fps. The table below reports the token‑length distribution of the text inputs; we will incorporate this information, along with additional dataset details, in the paper’s final draft.
>
>
> **Word count distribution of commentary text:**
>
> | **Metric**           | **Correct Commentary** | **Distractor Commentary** |
> |----------------------|:------------------------:|:----------------------------:|
> | Mean                 |    25.00                  |    25.03                      |
> | Median               |    24.00                  |    24.00                      |
> | Std Dev              |    9.66                   |    8.33                       |
> | 25th Percentile      |    19.00                  |    20.00                      |
> | 75th Percentile      |    30.00                  |    28.00                      |
>
>
> **Limited diversity of activity categories.**
>
> Our benchmark is built on top of the Ego-Exo4D dataset—a massive, multi-year effort involving millions of dollars in funding and contributions from hundreds of researchers across institutions. The core objective of Ego-Exo4D was to produce a large-scale, high-quality, and diverse dataset of skilled human activities, with expert annotations and coverage across multiple domains. Building on this, our work introduces a formal benchmark for expert action analysis, formulating it as a multiple-choice question-answering task, which enables easy and rigorous evaluation.
>
> Importantly, as also noted by Reviewer Uwg9, the combination of domain diversity and expert-level QA in our benchmark is not present in prior benchmarks such as FineGym, BASKET, or ActivityNet, which are typically limited to either broader general activities that do not require expert-level understanding (ActivityNet) or very few domains (e.g., 1-2) requiring expert-level knowledge (BASKET, FineGym). In contrast, our ExAct spans 6 distinct expert-level domains, including Sports, Cooking, Music, Health, and others, and covers 11 expert-level activities.
>
> While we agree that expanding action diversity further would be valuable, even assembling the current set of expert-level annotations required extensive effort and resources. We plan to expand to more domains and activities in future iterations to further broaden the scope and diversity of expert-level skill analysis.
>
> **Are evaluation questions sufficiently fine-grained for expert-level assessment?**
>
> The evaluation questions in our benchmark are designed to capture expert-level nuances, where subtle differences in technique and execution are critical. Each question is carefully reviewed by domain experts to ensure it captures fine-grained, expert-level distinctions with clarity and accuracy. The performance gap between domain experts (82% accuracy) and non-experts (60% accuracy) further demonstrates that our QA pairs require specialized knowledge and are effective at distinguishing expert-level understanding from general familiarity. This indicates that the questions are sufficiently fine-grained across diverse domains. We encourage interested reviewers to examine the samples directly. The dataset is publicly available at the link previously provided in the submission.

---

### Official Review · Reviewer_mMq8 · 2025-07-03

**Rating:** 4
**Confidence:** 4

**Summary:**

The paper proposed dataset containing expert actions across diverse domains from sports to daily activities and musical instrument performance. It mainly support the multiple choice question answering task.

**Additional Feedback:**

Overall, I expect either novel tasks or challenging annotations rather than just identification questions at a premium venue like NeurIPS. I am leaning on negative rate for this paper, but open to discuss its potentials with other reviewers and the authors.

**Dataset Code Accessibility:**

Yes

**Dataset Code Comments:**

Links are given in the paper and sufficiently accessible, though I have not tried to run.

**Ethical Considerations:**

No, there are no or only very minor ethics concerns

**Final Justification:**

The author addressed my concerns adequately, I raise my rating.

**Limitations Weaknesses:**

- While the details in expert actions are presented in the dataset, only a single task which is QA is not impressive. Other recommended tasks can be temporal grounding to specify timestamp of the event, or more types of questions, such as causal and temporal-related questions as similar as NExT-QA or IntentQA, rather than just identification questions. Annotations support R1-like thinking process to teach the model differentiate between expert and non-expert actions would be a big plus if it has.

- How do the finetuned models on the training set perform? I would love to see whether it the dataset is easy to overfit.

- [Minor] The data may raise a concern on containing non-consented faces (though they are small).

**Strengths Contributions:**

- The proposed annotation of expert action sounds interesting and well-motivated.

- Clear in presentation

---

> ### Author Rebuttal · Authors · 2025-07-30
>
> **Task diversity and required capabilities.**
>
> While our questions follow a consistent template of selecting the best expert commentary for a given video, answering them requires a broad range of reasoning capabilities beyond simple identification. The evaluated capabilities include:
>
>
>
> * Fine-grained recognition: Detecting subtle differences in body positioning, movement patterns, and technique execution (Examples: "plays with a very tight wrist, ensuring all strings are treated equally" vs. "uses a hybrid picking technique, ensuring all strings are plucked independently.")
> * Temporal reasoning: Understanding timing precision, action phases/sequence, and temporal dependencies (Examples: "releases the ball too early in the shooting motion" vs. "releases the ball too late after the peak jump.")
> * Causal understanding: Linking technique flaws to performance outcomes through cause-and-effect reasoning. (Examples: "doesn't bend knees...reduces jump height" vs. "doesn't bend knees...prevents ball spin.")
> * Procedural understanding: Evaluating adherence to established protocols and sequential steps in structured activities. (Examples: "The participant should prioritize checking the victim's pulse and breathing before calling for help" vs. "The participant should delay calling for help and requesting an AED.")
> * Spatial & geometric reasoning: Understanding angles, trajectories, and spatial relationships critical to performance. (Examples: "decrease power and flatten the arc on their left-handed layup" vs. "lower the ball's angle of entry to avoid extensive rotation.")
> * Domain-specific expertise: Requiring professional-level knowledge across sports, music, cooking, healthcare, bike repair, and dance. (Examples: "positioning her comfortably within the recommended range (100 to 120 bpm), which is crucial for effective CPR." vs. "positioning her comfortably within the recommended range (120 to 140 bpm), which is crucial for effective CPR.")
> * Physical understanding: Analyzing how biomechanical principles, force generation, and energy transfer affect movement execution. (Examples: "angle their knees inward while shooting to generate more power" vs. "opening their stance more instead of pointing directly at the rim") \
>
>
> Creating this benchmark required close collaboration with professional coaches and certified experts, including professional athletes and instructors across six distinct domains. Our novel benchmark enables a rigorous evaluation of expert-level physical skill understanding, which was previously not possible using the existing annotations of Ego-Exo4D (due to the annotations' format and the lack of a formal evaluation protocol).
>
> While prior works like TemporalBench, NExT-QA, and IntentQA explore similar QA formats, they do not target fine-grained, expert-level understanding of physical human skills. As Reviewer Uwg9 noted, existing benchmarks often trade off between diversity and depth—datasets like ActivityNet span multiple domains but lack expert-level reasoning, while others like FineGym and BASKET are fine-grained but limited in scope (1-2 domains). Our benchmark fills this critical gap by combining expert-level, fine-grained analysis with broad domain coverage across multiple real-world activities. This enables more meaningful evaluation of models intended for real-world applications such as automated coaching, training, and skill assessment.
>
> **Finetuning experiments.**
>
> First, we would like to clarify that the primary objective of our dataset is to serve as a zero-shot evaluation benchmark for testing state-of-the-art VLMs. Since our dataset contains only a few thousand annotated samples, we don't have a designated training set, and we use all of these samples for evaluation. However, to address the reviewer’s request, we conducted several finetuning experiments for the rebuttal.
>
> Specifically, to finetune the models, we dedicated 1,000 samples to be used as a training set while the remaining 2,521 samples were used as a validation set. We used LLaVA-Video 7B model and followed their standard configuration to finetune their model on this training set. After finetuning, we report that the model produced a validation accuracy of 43.03%, which is slightly higher than the best zero-shot open-source model (41.58%) but still significantly lower than both non-expert human performance (61.86%) and expert human performance (82.02%). We will include these experiments in the final draft of our paper.
>
> **Privacy concerns.**
>
> Our benchmark dataset is built upon the Ego-Exo4D dataset. As explicitly stated in their paper (page 54), "All participants consented to their data being collected and distributed for research purposes," ensuring all data usage complies with consent requirements. Therefore, we do not anticipate any issues related to participant consent.

---

> ### Author Response · Authors · 2025-08-05
>
> Dear Reviewer mMq8,
>
> Since the rebuttal period is ending soon, we wanted to politely check in to see if you had the chance to review our response, which hopefully address all your questions. We would be happy to further discuss any questions or suggestions you might have, and we hope you might consider revisiting your rating in light of these new positive findings.
>
> Best,
>
> Authors

---

### Official Review · Reviewer_p76Z · 2025-07-03

**Rating:** 4
**Confidence:** 4

**Summary:**

The paper introduces EXACT, a video-language benchmark designed to assess expert-level understanding of skilled activities. The dataset includes 3,521 multiple-choice video Q&As across 11 activities in 6 domains, including Sports, Bike Repair, Cooking, Health, Music, and Dance. EXACT aims to drive progress in building AI models that can provide accurate, expert-like feedback on human actions. Existing vision-language models (VLMs), including GPT-4o and Gemini, perform poorly on EXACT compared to human experts. This benchmark highlights a significant gap between current VLMs and expert-level human understanding.

**Dataset Code Accessibility:**

Yes

**Dataset Code Comments:**

The paper provides the benchmark dataset with its Huggingface host link and the evaluation code on GitHub.

**Ethical Considerations:**

No, there are no or only very minor ethics concerns

**Final Justification:**

The authors’ response addressed my concerns. After considering other reviews and responses, I have decided to raise my rating to borderline accept, given the benchmark’s focus on advancing expert-level understanding of skilled activities.

**Limitations Weaknesses:**

- The benchmark is built on top of the Ego-Exo4D dataset, which limits the action diversity.
- To reduce the bias of the length of each choice, the authors decided to make all choices of similar length (absolute word count difference < 8), which limits the diversity of choices.
- Why are human experts only achieving around 80% accuracy? Are some of the generated choices too similar or ambiguous to distinguish by humans? Or are the questions too hard so that the human experts are not able to identify the correct solutions?
- L280: If mid-level resolution shows the best performance, why don't the authors stick with the mid-resolution for all evaluations?

**Strengths Contributions:**

- This paper introduces a multiple-choice questions benchmark focused on expert-level understanding of skilled human activities, which is lacking in existing video-language datasets.
- The authors apply a 4-stage annotation process, including expert validation and bias filtering, to ensure high-quality multiple-choice questions.
- The paper evaluates a wide range of state-of-the-art VLMs, revealing clear performance gaps compared to humans.

---

> ### Author Rebuttal · Authors · 2025-07-30
>
> **Limited action diversity.**
>
> Our benchmark is built on top of the Ego-Exo4D dataset—a massive, multi-year effort involving millions of dollars in funding and contributions from hundreds of researchers across institutions. The core objective of Ego-Exo4D was to produce a large-scale, high-quality, and diverse dataset of skilled human activities, with expert annotations and coverage across multiple domains. Building on this, our work introduces a formal benchmark for expert action analysis, formulating it as a multiple-choice question-answering task, which enables easy and rigorous evaluation.
>
> Importantly, as also noted by Reviewer Uwg9, the combination of domain diversity and expert-level QA in our benchmark is not present in prior benchmarks such as FineGym, BASKET, or ActivityNet, which are typically limited to either broader general activities that do not require expert-level understanding (ActivityNet) or very few domains (e.g., 1-2) requiring expert-level knowledge (BASKET, FineGym). In contrast, our ExAct spans 6 distinct expert-level domains, including Sports, Cooking, Music, Health, and others, and covers 11 expert-level activities.
>
> While we agree that expanding action diversity further would be valuable, even assembling the current set of expert-level annotations required extensive effort and resources. We plan to expand to more domains and activities in future iterations to further broaden the scope and diversity of expert-level skill analysis.
>
> **Candidate answer length similarity does not limit the diversity of answer choices.**
>
> In our experiments, we observed that significant length differences among options can introduce unintended biases, making it easier for both humans and models to identify the correct answer based on superficial cues rather than genuine video/skill understanding. Prior to adopting our current constraints, we experimented with varying the length of answer candidates. However, expert annotators consistently reported that noticeable length discrepancies made it easier to spot patterns, encouraging superficial judgments rather than careful reasoning. To mitigate such bias, we applied an empirical constraint: all distractors must be between 80%–120% of the correct commentary’s length, and the absolute word count difference must not exceed 8 words. This range was chosen based on annotator feedback and the layout of our annotation interface, where an 8-word difference typically corresponds to half a line, making length-based discrimination less likely. Additionally, we provide the full word count distribution of commentary texts in the following table to transparently show the natural variation within the dataset.
>
>
> **Word count distribution of commentary text:**
>
> | **Metric**           | **Correct Commentary** | **Distractor Commentary** |
> |----------------------|:------------------------:|:----------------------------:|
> | Mean                 |    25.00                  |    25.03                      |
> | Median               |    24.00                  |    24.00                      |
> | Std Dev              |    9.66                   |    8.33                       |
> | 25th Percentile      |    19.00                  |    20.00                      |
> | 75th Percentile      |    30.00                  |    28.00                      |
>
>
>
> **Human expert accuracy is around 80%.**
>
> It's important to consider the complexity of the questions in ExAct. These questions require not only domain-specific knowledge but also careful attention to subtle human actions and their consequences. Since the dataset requires subtle, fine-grained, expert-level recognition, we observe that even experienced annotators may sometimes miss subtle cues, leading to errors. It's also important to contextualize the ~82% expert accuracy. Non-expert human annotators typically perform around 60%, so a 22-point gap reflects a meaningful advantage from domain expertise. Given the inherent complexity of expert-level skill assessment, we believe that 82% accuracy is reasonable and appropriate for this challenging benchmark.
>
> **Why not use mid-resolution for evaluations?**
>
> Higher-resolution input is critical for skill assessment, as it provides more visual detail necessary for capturing subtle, nuanced, and fine-grained human actions. This was also confirmed by our human annotators, who were provided the same high-resolution input videos. Motivated by this feedback from annotators, and to ensure fairness across all model comparisons, we standardized the input resolution to 796 × 448 for all our tested models.

---

> > ### Comment · Reviewer_p76Z · 2025-08-06
> >
> > Thank the authors for their detailed response. It has addressed my concerns.

---

> ### Author Response · Authors · 2025-08-05
>
> Dear Reviewer p76Z,
>
> Since the rebuttal period is ending soon, we wanted to politely check in to see if you had the chance to review our response, which hopefully address all your questions. We would be happy to further discuss any questions or suggestions you might have, and we hope you might consider revisiting your rating in light of these new positive findings.
>
> Best,
>
> Authors

---

### Official Review · Reviewer_Uwg9 · 2025-07-03

**Rating:** 5
**Confidence:** 4

**Summary:**

ExAct introduces a new video benchmark for expert action analysis. This task can be thought of as a qualitative version of the more common action quality analysis task. The paper’s primary contributions are the benchmark itself and the finding that VLMs struggle on expert-level video tasks.

**Additional Feedback:**

- Line 10: “trained human specialists/experts” may be interpreted to mean that you have trained the experts. Please make this wording more clear.
- Figure 4(a): capitalize the ‘s’ in “COVID-19 safety” to make it consistent with the other x-axis labels and the capitalization in Figure 2.

**Dataset Code Accessibility:**

Yes

**Dataset Code Comments:**

Yes. Evaluation code is provided in the linked GitHub repository, and the benchmark data is provided in the linked Hugging Face collection.

**Ethical Considerations:**

No, there are no or only very minor ethics concerns

**Final Justification:**

Thank you for the response. The additional data annotation details are helpful and authors additional report performance of the latest SoTA properitary and open-osurced models (which authors should include in the final version), showing the gap with human performance. I maintain my rating.

**Limitations Weaknesses:**

- Figure 2 makes evident how uneven the distribution of samples across categories is in this benchmark. In particular, basketball has significantly more samples than any other category. Given that the paper positions itself as more diverse than existing benchmarks, it would be prudent to justify why this is not an issue. Even better, it could be useful to have a section in the appendix that considers a different overall evaluation metric (which adjusts for the uneven distribution) to see if that changes relative model performance.
- In the related works table, much attention is given to “free-form language annotation” and “MCQ evaluation.” It is not clear why these are necessary parts of a good benchmark and why the alternative annotation and evaluation techniques used by other benchmarks are worse. That being said, the benchmark is sufficiently different from existing benchmarks in its task (expert action analysis over a diversity of categories) to make up for this omission. There is limited attention provided to it in illness 41-44; please elaborate on these ideas.
- Lines 83-86 make claims that are unsupported by evidence (although the reviewer agrees that the claims are true per existing literature). Please add a few cites here.
- Lines 110-112 discuss Ego-Exo4D and its downsides. These downsides are mentioned again later in the paper. It could be helpful to add some indicators of how common these pitfalls are, so the audience understands how important (or un-important) some steps of your benchmark pipeline are.
- Line 119 suggests that GPT-4 corrects errors in the commentaries (as in the commentary is inaccurate to the video or is inaccurate to what is generally believed to be true in an expert community), but Subsection 3.1 indicates that GPT-4 only corrects transcription errors. Please make clear in the paper which one of these two interpretations is correct.
- In Subsection 3.2, only two strategies are used for generative negative commentaries for good execution. While these strategies seem strong, please elaborate on why it is not problematic to use such few heuristics when designing distractors. It is tempting for the reader to wonder if the distractors are too systematic in nature (and hence may be “cheated” by a VLM). Additional evidence supporting how difficult the generated negative commentaries are, or what experts believe of these commentaries, may also be beneficial here.
- Please elaborate on how you decided on the heuristics in lines 168-169, specifically the 80-120% threshold and the 8 word threshold. If this is inspired by previous works please cite accordingly.
- On lines 174 and 175, please elaborate on which LLMs were used for the Blind-LLM filtering. Please indicate why those LLMs were selected. Also please indicate if one response was generated per model, or if the same model was queried multiple times. Overall just more details here would be great.
- The justification for Stage III is strong, but there is a lack of evidence that the filtering strategies actually do what they intend to. Some statistics here would be great.
- Line 185 suggests that each QA sample is reviewed by “multiple” domain experts, but line 199 suggests that some samples are reviewed by only one expert. Please be clear about how many experts reviewed each sample. If this number varied across, please indicate what the mean/median/etc. Was.
- Do the experts tend to agree with each other? Many actions have multiple interpretations of how they should be executed vs. how they should not be executed. It could be worthwhile to include a section in the appendix which addresses why we should trust the experts (beyond their years of experience) and if their thoughts on samples are consistent or not.
Please include statistics on the video length so the reader can get a feel for what effects uniformly sampling 32 frames (as discussed on line 218) may have.
- The extremely recent Perception-LM is tested, but none of the more recent Gemini (2.0 Flash, 2.5 Pro, etc.) models or GPT-4.1 were tested. While it is clear from the existing tests that proprietary models struggle, Gemini 2.5 Pro and GPT-4.1 have beaten Gemini 1.5 Pro and GPT-4o on other benchmarks, so it may be prudent to test those models to establish an upper-bound on current SoTA model performance for the benchmark. This would also provide a good goal for future works that may try to improve open models on the benchmark.
- Line 245: please describe non-experts more. Have they been trained by you at all, or at least filtered / selected to be “good workers” based on some pre-annotation task, or are they truly random crowdworkers? This provides important context to the non-expert statistics in Table 2.
- What would Frequency Choice be in Table 2? If it is significantly different from Random Choice then it may be good to include.
For table 3(b), it may be good to use a model with a 32B variant so there are 3 data points instead of 2, making it easier to map out a trend.
- For the prompting strategies paragraph (lines 274-278), the reviewer recalls the documentation for LLaVA recommending time instructions in a similar format. If this is the case, it is not clear if the 1.5% accuracy gain comes from a better temporal understanding or the format of the questions being more in-distribution for the VLM.

**Strengths Contributions:**

- The greatest strength of the paper is how unique the benchmark is compared to existing literature. It seems that there are diverse benchmarks (e.g., ActivityNet) that cover a variety of domains, but such benchmarks fail to be fine-grained or require expert-level knowledge. On the other hand, there are a few benchmarks that are fine-grained and require expert-level knowledge (e.g., FineGym or BASKET), but they are limited to one or two domains. The combination of diversity and fine-grained/expert-level questions presented in this paper is a unique and meaningful contribution.
- The paper provides a robust pipeline for generating distractors and verifying that they are not easy to cheat. In particular, the strategies described in Subsection 3.2 are subtle and akin to how a human might approach the problem, while the strategies in Subsection 3.3 appear to eliminate most of the “cheatable” questions. This culminates in MCQ evaluation, which is much less prone to answer extraction problems.
- The benchmark is validated by domain experts, providing a high level of confidence in its accuracy / validity.
- The paper does a great job motivating its real-world applicability in the introduction.
- There is great analysis in Section 5.1 of the types of domains in which models struggle and the types of domains in which they excel, along with why this may be the case.
- Paper is well-written and the figures are clean.

---

> ### Author Rebuttal · Authors · 2025-07-31
>
> **Uneven sample distribution across categories.**
>
> The original Ego-Exo4D dataset contains substantial variation in the number of samples across domains, with Cooking and Basketball having the largest amounts of data. Our filtering pipeline was designed to ensure data quality, rather than altering domain distributions. As a result, the existing imbalance persists.
>
> To mitigate this imbalance, we report per-domain results in the main table (Table 2), enabling a clear comparison of model performance across individual domains.
>
> Additionally, as requested by the reviewer, we include a macro-averaged accuracy metric to adjust for category imbalance. Unlike micro-average accuracy, which aggregates predictions across all samples, macro-average accuracy computes the accuracy within each domain and then takes the unweighted average across domains, giving equal importance to each domain. We report these results below and note that the overall trend remains largely unchanged, suggesting that performance differences are relatively consistent across domains. We will include both metrics in the final draft.
>
> |**Acc. (%)**|PerceptionLM|VideoLLaMA3|InternVL2.5|LLaVA-OneVision|Qwen2.5-VL|LLaVA-Video|Gemini 1.5 Pro|GPT-4o|Human Non-Expert|Human Expert|
> |-|-|-|-|-|-|-|-|-|-|-|
> |Micro-average accuracy (original one)|24.65|26.38|33.48|35.44|35.67|41.58|43.91|44.70|61.86|82.02|
> |Macro-average accuracy|25.01|26.25|34.46|36.60|35.81|41.39|44.95|46.07|62.41|82.07|
>
> **Why are free-form annotation and MCQ evaluation important?**
>
> Free-form language enables more expressive, and open-ended analysis of skilled actions, capturing subtle errors, temporal nuances, and intent—details that scalar or categorical labels often miss. Conversely, the MCQ format eliminates the ambiguity and reproducibility issues associated with open-ended evaluation by offering clearly defined answers and allowing control over question difficulty and the design of distractor answers.
>
> **The importance of data filtering.**
>
> The original Ego-Exo4D dataset contains over 400,000 expert commentaries. After applying our filtering pipeline, which includes ASR correction, removal of irrelevant content, and strict quality control, fewer than 110,000 remained. The noisy nature of Ego-Exo4D commentaries underscores the importance of preprocessing steps in our design.
>
> **Negative commentary generation strategies (Subsection 3.2).**
>
> While we use only two strategies to generate negative commentaries, our pipeline is deliberately general and designed to probe a wide range of capabilities. As discussed in our response to reviewer mMq8, our generated questions require the following capabilities: 1) fine-grained recognition, 2) temporal reasoning, 3) causal understanding, 4) procedural understanding, 5) spatial & geometric reasoning, 6) domain-specific expertise, and 7) physical understanding. This design enables the construction of diverse and challenging distractors that effectively support our comprehensive evaluation.
>
> Furthermore, to prevent models from exploiting superficial cues, we applied blind-LLM filtering, discarding samples where models could consistently guess the correct answer. This removed over 90% of the initial samples, ensuring the remaining ones require genuine visual reasoning. Domain experts then manually verified the plausibility and difficulty of the retained distractors. Experts confirmed that these negative commentaries were realistic, non-trivial, and sufficient to challenge both models and non-expert humans.
>
> **Justification for length-based filtering in Lines 168-169.**
>
> The 80–120% length range and 8-word absolute difference threshold were determined empirically through iterative pilot studies with human annotators. Annotators reported that noticeable length differences between answer choices often drew attention to superficial patterns over content.
>
> We also considered the visual layout of our annotation interface, where an 8-word gap typically translates to roughly half a line of text, which annotators found visually salient. By applying this threshold, we aimed to minimize length-based biases without overly restricting the natural variation in commentary phrasing. To the best of our knowledge, these specific thresholds are not directly drawn from prior work but were selected based on practical constraints and annotator feedback in our setting. We will clarify this in the paper.
>
> **Blind-LLM filtering.**
>
> In the Blind-LLM filtering stage (Stage III in Figure 3), we used five strong LLMs: GPT-4o and DeepSeek-VL-R1 (proprietary), Qwen2.5-72B, LLaMA3.3-70B, and InternLM2.5-20B (open-source, all ranked highly on multiple LLM leaderboards). These models were selected for their strong performance on general-purpose reasoning benchmarks. We provided only textual inputs at this filtering stage. For each sample, we generated one response per model. We will clarify these details in our revised draft.
>
> **Sample statistics after each filtering stage.**
>
> Starting from >400k expert commentaries in Ego-Exo4D, Stage I reduced the number of samples to 108k (≈ −73%). Stage III's length similarity filtering further reduced it from 108k to 60k (≈ −44%). The blind-LLM filtering further shrank the sample size from 60k to 4.5k (≈ −92.5%). Finally, Stage IV expert verification led to 3.5k samples (≈ −22% of the 4.5k), leaving ~0.9% of the original samples. These statistics demonstrate that Stage III is critical as it filters low-quality or easily "cheatable" samples before sending them to the experts for final verification.
>
> **Clarification on the number of expert reviewers per sample (Lines 185 vs. 199).**
>
> Across the 11 activity domains, we recruited 16 domain experts, each with over 10 years of experience in their respective fields. We provided high compensation to ensure sustained engagement and high annotation quality.
>
> To balance annotation workload with expert availability, we adopted the following protocol: for five higher-volume domains—basketball, bouldering, cooking, dance, and bike repair—each sample was reviewed by two experts and retained only if both independently approved it. For the remaining six domains, each sample was reviewed by one qualified expert.
>
> This setup is consistent with prior expert-annotated datasets such as Ego-Exo4D, where expert commentaries were typically reviewed and written by a single domain expert.
>
> **Expert agreement.**
>
> In our benchmark, all multiple-choice questions were carefully constructed to include exactly one correct answer and four incorrect distractors. This design inherently reduces ambiguity, as the goal is not to elicit open-ended interpretation but to identify the objectively correct commentary. For domains reviewed by two experts, we observed that experts reached agreement in over 90% of cases—demonstrating strong consistency in their judgments.
>
> **Statistics on the video length.**
>
> Statistics on the video length used in our benchmark are shown in Figure 2 (top right) of the main draft. Across all 3,521 QA samples, the videos have an average duration of 13.76 seconds, with a standard deviation of 13.09 seconds. The shortest video is 2.57 seconds, and the longest is 54 seconds. The distribution is right-skewed: 50% of videos are under 7 seconds, and 75% under 18 seconds.
>
> **Results of newer models.**
>
> We have conducted additional evaluations using GPT-4.1 and Gemini 2.5 Pro. Notably, even these state-of-the-art models exhibit a huge gap compared to human expert performance, indicating that current models are not yet capable of reliably solving this challenging task.
> |Model|Overall|Sports|Bike Repair|Cooking|Health|Music|Dance|
> |-|-|-|-|-|-|-|-|
> |GPT-4.1|50.89%|51.37%|58.90%|54.25%|51.10%|40.84%|51.48%|
> |Gemini-2.5 pro|55.36%|52.58%|65.05%|58.36%|60.71%|53.05%|53.98%|
> |Human Non-Expert|61.86%|62.97%|55.02%|66.58%|71.43%|59.22%|59.22%|
> |Human Expert|82.02%|82.09%|81.23%|80.27%|87.09%|80.21%|81.55%|
>
> **Non-expert evaluation.**
>
> Non-expert subjects were carefully selected university students who had demonstrated reliability in prior research studies. All participants completed pilot rounds with detailed task guidelines to ensure they understood the evaluation format. No random crowdworkers were used to ensure the high quality of our study.
>
> **Frequency baseline.**
>
> We randomly shuffled the answer choices to ensure that no option consistently appears in a particular position across questions. As a result, a Frequency Choice baseline yields 20% accuracy.
>
> **Adding a 32B model.**
>
> We added results for Qwen2.5-VL-32B, enabling comparison across the 7B, 32B, and 72B variants. As shown in the results below, accuracy consistently improves with model size.
> |Model Size|Acc. (%)|
> |-|-|
> |7B|31.85|
> |32B|33.83|
> |72B|35.67|
>
> **Prompt format.**
>
> We conducted an additional ablation using a prompt with a markedly different temporal instruction format. In this variant, we replaced the original natural-language phrasing with a more structured, synthetic format:
>
> * Original prompt: \
> "The video is {video_time}s long, and {len(frames)} uniformly sampled frames occur at {frame_time}."
> * New prompt: \
> "TEMPORAL CONTEXT: Duration={video_time:.2f}s | Frame_count={len(video[0])} | Timestamps={frame_time}"
>
> The baseline accuracy without any time information in the prompt is 40.04%. Using the original prompt with time information improves it to 41.58%, while the new prompt achieves 42.26%. This suggests that the gains come from the inclusion of explicit time information, rather than the prompt format.
>
> **Clarification of "errors" in Line 119 vs. Subsection 3.1.**
>
> Both Line 119 and Subsection 3.1 refer to the same process: using GPT-4o to correct ASR errors, remove irrelevant or repetitive content, and segment transcripts into concise expert commentaries. These are linguistic issues, unrelated to the video content or the correctness of expert feedback itself.
>
> **Cite paper in Lines 83-86.**
>
> We will cite the relevant papers in the revised draft.

---

> > ### Comment · Reviewer_Uwg9 · 2025-08-07
> >
> > Thank you for the response. The additional data annotation details are helpful and authors additional report performance of the latest SoTA properitary and open-osurced models (which authors should include in the final version), showing the gap with human performance. I maintain my rating.

---

### Note · Authors · 2025-08-12

**We thank the reviewers and AC for their time and constructive feedback. We believe we have addressed all concerns raised during the review process.**



* Reviewer Uwg9: We provided additional details on our data filtering and annotation process, and reported new results for the latest state-of-the-art proprietary and open-source models, further illustrating the substantial performance gap between current models and humans. We also explained the importance of both free-form annotation and MCQ evaluation, and addressed concerns in our ablation study by adding results for a 32B model and testing a markedly different temporal instruction format.
* Reviewer p76Z: We addressed concerns about dataset diversity and explained why human expert accuracy is around 80%, highlighting the challenging, fine-grained, and domain-specific nature of the tasks.
* Reviewer mMq8: We detailed the task diversity and the capabilities required by our benchmark, and conducted fine-tuning experiments. After fine-tuning, the model’s performance improved only modestly and remained far below human levels, underscoring the benchmark’s difficulty and the substantial gap between current models and expert understanding.
* Reviewer PK3K: We provided comprehensive details on the human annotation process, addressed concerns about dataset diversity, and clarified that our evaluation questions are sufficiently fine-grained to enable expert-level skill assessment.

Overall, we believe the revisions and additional analyses have resolved all reviewer concerns, and we look forward to improving the paper further in the camera-ready version.

---

### Decision · Program_Chairs · 2025-09-18

**Decision:**

Accept (poster)

**Comment:**

This paper proposes a video–language benchmark targeting expert-level understanding of skilled human activities. The reviewers generally agree on the paper combines diversity across domains with fine-grained, expert-level questions, filling a gap left by prior datasets. The evaluation reveals a pronounced model–human gap and includes informative analyses of where models struggle and excel.

In the final assessment, this paper is technically correct and well executed, but the constraints on the number of papers that can be accepted at the conference necessitates more scrutiny into the significance of the contribution. The reviewers generally expressed reservations about the breadth of the benchmark, and a number of questions raised about potential issues. While none of these issues seem catastrophic, and the benchmark could well be useful to the community, the constraints on the number of papers that can be accepted into the D&B track required us to judge the work not just on its correctness but also on scope and potential for impact, which we felt would not be as high as other benchmarks. This was a difficult decision, because it's clear that the benchmark can provide value to the community, but compared to other papers we felt that it is not currently above the bar for NeurIPS.

===== FINAL UPDATE FROM DB Track PCs ====

The final decision for this paper has been taken by the program chairs after consultation with the SACs. All Senior Area Chairs have ranked papers according to the feedback from the AC during the review process. We decided to leave the original meta-review to reflect the opinion of the AC in light of the initial discussions with reviewers and SAC.